



# Performance evaluation of the Alphasense OPC-N3 and
# Plantower PMS5003 sensor in measuring dust events in the Salt
# Lake Valley, Utah
Kamaljeet Kaur[1], Kerry E. Kelly[1]
[1]Department of Chemical Engineering, University of Utah, SLC, 84102, USA
*Correspondence to*: Kerry Kelly (kerry.kelly@utah.edu)
**Abstract.** As the changing climate expands the extent of arid and semi-arid lands, the number, severity of, and health
effects associated with dust events are likely to increase. However, regulatory measurements capable of capturing dust
($PM_{10}$, particulate matter smaller than 10 µm in diameter) are sparse, sparser than measurements of $PM_{2.5}$ (PM smaller
than 2.5 µm in diameter). Although low-cost sensors could supplement regulatory monitors, as numerous studies have
shown for $PM_{2.5}$ concentration, most of these sensors are not effective at measuring $PM_{10}$ despite claims by sensor
manufacturers. This study focuses on the Salt Lake Valley, adjacent to the Great Salt Lake, which recently reached
historic lows exposing 1865 km² of dry lakebed. It evaluated the field performance of the Plantower PMS 5003, a
common low-cost PM sensor, and the Alphasense OPC-N3, a promising candidate for low-cost measurement of $PM_{10}$,
against a federal equivalent method (FEM, beta attenuation) and research measurements (GRIMM aerosol
spectrometer model 1.109) at three different locations. During a month-long field study that included five dust events
in the Salt Lake Valley with $PM_{10}$ concentrations reaching 311 µg/m³, the OPC-N3 exhibited strong correlation with
FEM $PM_{10}$ measurements ($R^2 = 0.865$, RMSE = 12.4 µg/m³) and GRIMM ($R^2 = 0.937$, RMSE = 17.7 µg/m³). The
PMS sensor exhibited poor to moderate correlations ($R^2 < 0.49$, RMSE = 33-45 µg/m³) with reference/research
monitors and severely underestimated the $PM_{10}$ concentrations (slope <0.099) for $PM_{10}$. We also evaluated a PM-
ratio-based correction method to improve the estimated $PM_{10}$ concentration from PMS sensors. After applying this
method, PMS $PM_{10}$ concentrations correlated reasonably well with FEM measurements ($R^2 > 0.63$) and GRIMM
measurements ($R^2 > 0.76$), and the RMSE decreased to 15-25 µg/m³. Our results suggest that it may be possible to
obtain better resolved spatial estimates of $PM_{10}$ concentration using a combination of PMS sensors (often publicly
available in communities) and measurements of $PM_{2.5}$ and $PM_{10}$, such as those provided by FEMs, research-grade
instrumentation, or the OPC-N3.
**1 Introduction**
Our changing climate is expanding the extent of arid and semi-arid lands globally; these lands currently cover
approximately 1/3rd of the Earth's land surface (Williams et al., 2022; Huang et al., 2016). Recent studies suggest that
this expansion of arid lands is linked to increases in the number and severity of dust events (Clifford et al., 2019; Tong
et al., 2017; Ardon-Dryer and Kelley, 2022). Dust events can transport particulate matter (PM), particle-bound air
toxics, and allergens over thousands of kilometers (Goudie, 2014). The suspended PM affects regional climate by



impacting cloud formation, precipitation processes, and convection activity (Cai et al., 2021; Kumar et al., 2021;
Mallet et al., 2009). Dust events significantly affect the regional air quality (Chakravarty et al., 2021; Akinwumiju et
al., 2021; Liu et al., 2020), decrease atmospheric visibility (Jayaratne et al., 2011) and have adverse effects on human
health, including being linked to increased incidence of asthma, pneumonia, bronchitis, stroke, adverse birth outcomes,
influenza, meningitis, and valley fever (Dastoorpoor et al., 2018; Jones, 2020; Bogan et al., 2021; Soy, 2016; Trianti
et al., 2017; Diokhane et al., 2016; Schweitzer et al., 2018).

During dust events, the majority of PM is greater than 2.5 µm in diameter (Tam et al., 2012). Government
organizations, such as the World Health Organization (WHO), measure and/or provide guidelines for ambient $PM_{10}$
concentrations ($PM_{10}$, particles with aerodynamic diameter <10 µm). PM smaller than 10 µm in diameter is of
particular interest because it is inhalable. The WHO has set guidelines for 24-hour and annual average $PM_{10}$
concentration at 45 and 15 µg/m$^3$, respectively (WHO, 2022). One challenge with these 24-hour guidelines is that dust
events often last a few hours, and these events are obscured when reporting only the $PM_{10}$ 24-hour average or
comparing these averages to the 24-hour and guidelines (Ardon-Dryer and Kelley, 2022).

$PM_{10}$ concentrations tend to be more spatially heterogenous than $PM_{2.5}$ concentrations because $PM_{10}$ settles more
quickly (Keet et al., 2018). In addition, regulatory measurements of $PM_{10}$ are spatially and temporally sparser than
$PM_{2.5}$ measurements. For example, the US EPA reports measurements from 1,370 active $PM_{2.5}$ sites versus 800 active
$PM_{10}$ sites (EPA, 2022). Approximately half of these $PM_{10}$ sites only report 24-hour averages (USA EPA, 2022).
Furthermore, many dust-prone areas of the US lack any PM monitoring (USA EPA, 2022). More highly resolved
measurements of $PM_{10}$ concentration would aid communities and researchers in understanding and addressing the
effects of windblown dust and dust events.

More recent studies of PM have leveraged low-cost PM measurements and mobile measurements to obtain higher
spatial and temporal resolution $PM_{2.5}$ estimates (Bi et al., 2020; Caplin et al., 2019; Lim et al., 2019; Caubel et al.,
2019; Kelly et al., 2021). With appropriate calibration, low-cost sensors have been demonstrated to be generally
effective at measuring $PM_{2.5}$; however, the most common low-cost PM sensors that employ a laser, and a photodiode
to estimate particle concentration (Plantower PMS, Nova SDSS011, Sensirion SPS30, Shineyi PPD42NS, and
Samyoung DSM501A) are ineffective at measuring $PM_{10}$ and dust (Kosmopoulos et al., 2020; Mei et al., 2020; Sayahi
et al., 2019, Kuula et al. 2020) primarily due to the sensor's inability to aspirate these larger particles into the device
(Ouimette et al., 2022). Kuula et al. (2020) tested several low-cost PM sensors using monodisperse di-octyl sebacate
particles (0.5 – 10 µm) and observed a constant particle size distribution for particle sizes >0.5 µm and indicated that
these sensors are incapable of measuring coarse-mode particles (2.5-10 µm).

The Alphasense OPC-N series is a promising low-cost sensor for measuring $PM_{10}$. It is larger and more expensive
(~$500) than many of the low-cost PM sensors (<$50) with a greater flow rate (total flow of 5.5 LPM and sample
flow rate of 0.28 L/min) and a mirror that allows collection of light scattering from broader array of angles than typical



low-cost PM sensors, which have flow rates on the order of 0.1 LPM (Sayahi et al., 2019; Ouimette et al., 2022;
Alphasense Ltd, 2022). The OPC-N3 allows particle counting in 24-size bins for sizes ranging from 0.35-40 µm. The
working principle of Alphasense OPC-N3 and its previous version (OPC-N2) is similar to an aerosol spectrometer; it
measures scattering from single particles (Vogt et al., 2021). Studies have used the Alphasense OPCs for indoor and
ambient PM monitoring (Kaliszewski et al., 2020; Chu et al., 2021; Dubey et al., 2022b; Feenstra et al., 2019; Pope
et al., 2018; Nor et al., 2021; Alhasa et al., 2018; Mohd Nadzir et al., 2020), to monitor $PM_{2.5}$ personal exposure (Harr
et al., 2022a), to identify PM sources  (Harr et al., 2022b; Bousiotis et al., 2021), and to monitor occupational $PM_{2.5}$
and $PM_{10}$ exposure (Runström Eden et al., 2022; Bächler et al., 2020). The Alphasense OPCs correlate well ($R^2$ =
0.93-0.99) with $PM_{10}$ in laboratory studies (Sousan et al., 2021, 2016; Samad et al., 2021; Dubey et al., 2022a). The
field-based studies have reported somewhat lower correlations ($R^2$: 0.53 – 0.8) (Bílek et al., 2021; Dubey et al., 2022b,
a; Crilley et al., 2018), due to the variable ambient meteorological conditions and changing PM compositions. The
ambient PM ratios ($PM_{2.5}/PM_{10}$) in these previous studies were greater than 0.6, indicating the main contributions to
PM levels were from the fine PMs, rather than coarser PMs. The ratio of $PM_{2.5}/PM_{10}$ can provide crucial information
about particle origin and formation process (Xu et al., 2017; Speranza et al., 2014). Duvall et al. (2021) have suggested
evaluating the performance of $PM_{10}$ sensors for varying $PM_{2.5}/PM_{10}$ ratios, and dust events provide a great opportunity
to evaluate $PM_{10}$ sensor performance at ambient PM ratios <0.3.

Few studies have evaluated the performance of Alphasense OPCs for measuring $PM_{10}$ concentration during dust
events. Gomes et al. (2022) measured hourly $PM_{10}$ concentration exceeding 300 µg/m$^3$ using the OPC-N3 during
Saharan dust events in western Portugal. In Sarajevo, Bosnia-Herzegovina, Masic, et al. (2020) reported that for the
Aralkum Desert dust event, the OPC-N2 tracked GRIMM-11D $PM_{10}$ measurements but at a lower magnitude. Fewer
studies have compared the Alphasense OPCs with the regulatory monitors during dust events. Vogt et al. (2021)
reported that the OPC-N3 captures the long-range transported dust well, but slightly overestimates $PM_{10}$ concentration
(<120 µg/m$^3$) compared to a FIDAS (EN 16450 approved regulatory instrument). They also reported a moderate
correlation with $PM_{10}$ compared to FIDAS ($R^2$ = 0.58-064, and RMSE between 12-13 µg/m$^3$) and compared to a
gravimetric method ($R^2$ =0.71-0.74, and RMSE between 9-11 µg/m$^3$). Mukherjee et al. (2017) evaluated the OPC-N2
performance against a Met One beta attenuation monitor (BAM) over 12 weeks in the Cuyama Valley of California,
where PM concentrations are impacted by wind-blown dust events and regional transport; they reported a moderate
to good degree of correlation ($R^2$ = 0.53-0.81, depending on sampling orientation) for $PM_{10}$ (<750 µg/m$^3$). In general,
the studies report that the OPC-N2/N3 tracks the temporal variation of research/reference measurements but with
varying correlation factors.

A high $PM_{2.5}/PM_{10}$ ratio represents fine-dominated aerosols, likely corresponding to anthropogenic or other
combustion sources. Low ratios represent coarser particles (aerodynamic size between 2.5-10 µm) that tend to
correspond to wind-blown dust (Sugimoto et al., 2016). Sugimoto et al. (2016) classified aerosols as local dust when
the $PM_{2.5}/PM_{10}$ ratio was less than 0.1 and as transported dust when $PM_{2.5}/PM_{10}$ ratios were between 0.1 to 0.3. During
dust events, low-cost sensors like the Plantower PMSs can detect only a small portion of a particle size distribution,



and its response greatly depends on the particle size distribution and particle optical properties (Vogt et al., 2021).
This study explores the possibility of using a size-segregated correction factor ($PM_{2.5}/PM_{10}$ ratio) to infer $PM_{10}$
concentration from low-cost sensors that typically respond poorly to particles larger than 2.5 µm in diameter. If
successful, this technique could leverage the large number of existing low-cost sensor measurements that use the
Plantower PMS (and similar sensors) and improve spatial estimates of $PM_{10}$ concentration.

This study aims to evaluate the Alphasense OPC-N3 to complement common low-cost PM measurements to
understand $PM_{10}$ concentrations during dust events in the Salt Lake Valley. The Salt Lake Valley is particularly well
suited to studying dust events because it is affected by both regional dust events from the playas located to the west
of the valley and from the drying Great Salt Lake bed, which has reached historic lows with more than 1865 $km^2$ of
exposed lakebed (Perry et al., 2019). Under appropriate meteorological conditions, portions of this exposed lakebed
produce substantial dust plumes, and the winds can transport this dust directly into the populated areas of the Salt
Lake Valley (Perry et al., 2019).

**2 Methods**
This study focused on April of 2022 in the Salt Lake Valley, when it experienced five dust events (summarized in
Table 1). It relies on low-cost sensors and reference/research measurements at three different locations (Fig. 1): the
Utah Division of Air Quality (UDAQ)'s Hawthorne monitoring station (HW), the UDAQ's Environmental Quality
(EQ) station and surroundings, and a residential site (RS) in the northeast quadrant of the Salt Lake Valley. This period
included an hourly average FEM (Federal Equivalent Method) $PM_{10}$ concentration that reached 311 µg/$m^3$.





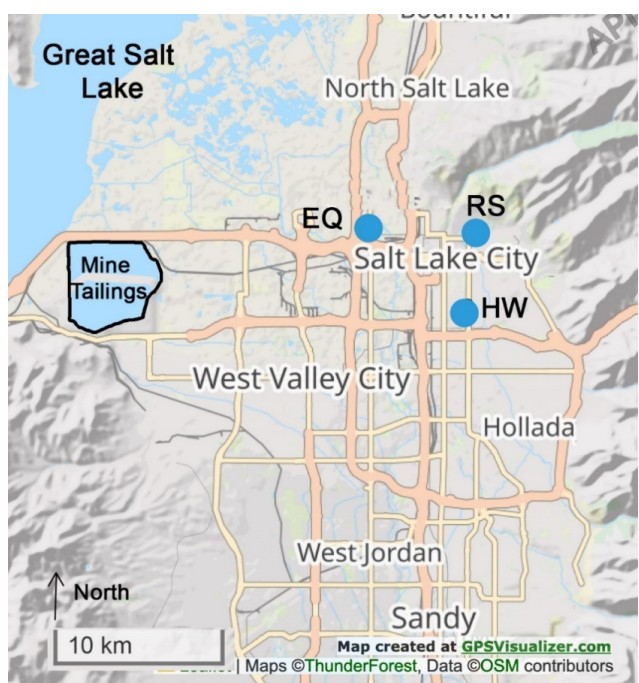

**Figure 1:** Study locations in Salt Lake County: EQ (UDAQ Environmental Quality) site, HW (Hawthorne UDAQ) site, and RS (residential site). The distance between EQ to HW, HW to RS, and EQ to RS is 7.8 km, 4.3 km, and 7.35 km, respectively. The OPC and PMS sensors were collocated at RS and HW sites. Two PurpleAir II were located within 2 km of the EQ monitoring station.





**Table 1:** PM measurements at the three different study locations.

| Site | Measurement type | Working principle | # | Sensor ID | Distance from a reference monitor | Hours of operation* |
|---|---|---|---|---|---|---|
| HW | OPC-N3 | Light Scattering (optical particle counter) | 1 | OPC-HW | Collocation | 633[a] |
| | PurpleAir II | Light Scattering-(nephelometry) | 2 | PMS-HW-1A, PMS-HW-1B, PMS-HW-2A, PMS-HW-2B | Collocation | 697 |
| | Thermo Scientific Model 5030 SHARP analyzer | Light scattering (nephelometry) + BAM | 1 | $PM_{2.5}$ FEM-HW | Federal equivalent method | 697 |
| | MetOne E-BAM PLUS | BAM | 1 | $PM_{10}$ FEM-HW | Federal equivalent method | 695 |
| EQ | PurpleAir II | Light Scattering-(nephelometry) | 2 | PMS-EQ-1A, PMS-EQ-1B, PMS-EQ-2A, PMS-EQ-2B | 480 m and 1.82 km | 697 |
| | Thermo Scientific Model 5030 SHARP analyzer | Light scattering (nephelometry) + BAM | | $PM_{2.5}$ FEM-EQ | Federal equivalent method | 697 |
| | MetOne E-BAM PLUS | BAM | | $PM_{10}$ FEM-EQ | Federal equivalent method | 697 |
| RS | OPC-N3 | Light Scattering (optical particle counter) | 1 | OPC-RS | Collocation | 425[c] |
| | PurpleAir II | Light Scattering-(nephelometry) | 2 | PMS-RS-1A, PMS-RS-1B, PMS-RS-2A, PMS-RS-2B | Collocation | 302[d] |
| | GRIMM 1.109 | Light Scattering (optical particle counter) | | GRIMM | Research monitor | 452 |

*Total number of hours = 711. Measurements corresponding to relative humidity >85%, i.e., 14 hrs, were
excluded.
[a]OPC-HW measurements were not available between 4/12/2022 6:00 pm – 4/14/2022 7:00 pm due to connectivity
issues.
[c]The measurements for OPC-RS were available starting 9 April 2022. OPC-RS measurements between 4/14/2022
10:00 am – 4/17/2022 20:00 pm were not available due to connectivity issues.
[d]The measurements from all the PurpleAir II at RS were available starting on 18 April 2022





### 2.1 Low-cost sensors

The low-cost sensors tested in this study include the Alphasense optical particle counter (OPC-N3, Alphasense Ltd, $500) and the Plantower PMS5003 ($20) integrated into the PurpleAir II (~$259). The Alphasense OPC-N3 uses a class 1 laser (~658 nm) to detect, size, and count particles in the size range 0.35-40 µm in 24 bins, which is translated, using the embedded algorithm, into estimated $PM_1$, $PM_{2.5}$, and $PM_{10}$ mass concentrations. The default setting for the OPC-N3's refractive index is 1.5 (real part) and for density is 1.65 $g/cm^3$, and these default settings were used throughout this study. The OPC-N3 uses an internal fan to create flow and reports a sample flow rate (~0.28 L/min and a total flow rate of 5.5 LPM). Each OPC-N3 was connected to a laptop and used the manufacturer-provided software. The OPC-N3 was set to store measurements every 1 min. The measurements included the date, size bins and counts, pump flow, relative humidity (RH), temperature, and $PM_1$, $PM_{2.5}$, and $PM_{10}$ concentration.

The PMS 5003 is a low-cost sensor (~$20, Plantower Technology, China), which has been integrated into a variety of low-cost air quality sensor packages, such as TSI BlueSky, PurpleAir, etc. It uses a fan to create a flow (~0.1 L/min), and it is equipped with a red laser (~680 ± 10 nm), a scattering angle of 90°, and a photo-diode detector to covert the scattered light to a voltage pulse (Sayahi et al., 2019; Ouimette et al. 2022). The PMS sensor converts light scattering into several different air quality parameters, including particle counts (0.3-10 µm), $PM_1$, $PM_{2.5}$, and $PM_{10}$, although these different metrics are all based on this single measurement, total light scattering. The PMS 5003 has been evaluated extensively in the laboratory and the field, and the measurements tend to correlate well with $PM_1$ or $PM_{2.5}$ concentration although it performs poorly for larger PM sizes, such as $PM_{2.5}$ - $PM_{10}$ (Sayahi et al., 2019; Vogt et al., 2021; Kuula et al., 2020; Ouimette et al., 2022). In this study, we used two PurpleAir PA-II at the HW and RS sites, each PA-II has two PMS sensors per node. $PM_{10}$ mass concentration corresponding to correction factor (CF) =1 and a data collection rate of every 2 minutes were used. The data were downloaded from the PurpleAir website. In addition, we evaluated two PurpleAir PA-II sensors located within 2 km of the EQ monitoring station.

All the OPC-N3 were placed inside a custom build housing to protect the sensor from rain and insects. The details of housing can be found in the supplementary material (Section S3).

### 2.2 Site descriptions

The study includes measurements from the two UDAQ sites (HW and EQ) in Salt Lake County that provide both hourly $PM_{2.5}$ and $PM_{10}$ measurements (Fig. 1). UDAQ uses a Thermo Scientific Model 5030 SHARP analyzer for measuring hourly $PM_{2.5}$ concentration and a MetOne E-BAM (Beta Attenuation Monitoring) PLUS for measuring $PM_{10}$ concentration. We placed two PurpleAir PA-II (containing four Plantower PMS 5003s, named: PMS-HW-1A, PMS-HW-1B, PMS-HW-2A, PMS-HW-2B) and one OPC-N3 (named: OPC-HW) at the HW site (Table 1). The PurpleAir PA-IIs and the OPC-N3 were mounted on poles that extend above the roof of the HW monitoring station. The HW monitoring station is located in an urban residential area (AQS: 49-035-3006, Lat: 40.7343, Long: -111.8721) at an elevation of 1308m. This site was established to represent population exposure in the Salt Lake City area, and it is often the controlling monitor for the county. The average of PMS-HW-1A, PMS-HW-2A, and PMS-HW-2B $PM_{10}$



concentrations at HW were named PMS-HW. PMS-HW-2B was excluded from the PMS-HW average because of its
moderate correlation with the other three sensors (Fig. S2).

We also evaluated two PurpleAir II (containing four Plantower PMS 5003s, named PMS-EQ-1A, PMS-EQ-1B, PMS-
EQ-2A, PMS-EQ-2B) sensors located near the UDAQ EQ site. One of the sensors was 480 m away (PMS-EQ-1),
while the other was 1.82 km away (PMS-EQ-2). The EQ monitoring station (AQS: 49-035-3015, Lat: 40.777028,
Long: -111.94585, elevation 1284 m) is located approximately 14 km southeast of the Great Salt Lake dry lake bed.
In addition to PM concentrations, we accessed relative humidity (RH), temperature, wind speed, and wind direction
data from the two UDAQ monitoring sites on EPA's AirNow Tech website. EPA-flagged measurements were
excluded from this study. UDAQ uses RM Young Ultrasonic Anemometer Model 86004 to measure the wind speed
and wind direction and an instrument based on a hygroscopic plastic film to measure relative humidity.

The RS was located in the northeast quadrant of the Salt Lake Valley at an elevation of 1383 m (40.771938, -
111.861290). Measurements at this site included four Plantower PMS 5003s (labeled as PMS-RS-1A, PMS-RS1B,
PMS-RS-2A, PMS-RS-2B) in two PurpleAir PA-IIs, one OPC-N3 (labeled as OPC-RS) and one GRIMM (model
1.109 Aerosol Technik Ainring, Germany). The GRIMM employs an internal pump to create a flow of 1.2 L/min and
measures the number concentration of particles of size 0.265 µm – 34 µm in 31 size bins, and reports estimated $PM_1$,
$PM_{2.5}$, and $PM_{10}$ concentrations. The GRIMM measurements were stored every minute in an internal storage card.
The GRIMM measurements were not available between 4/24/2022 6:00PM -4/26/2022 2:00 PM MDT (Mountain Day
Time). The PurpleAir PA-IIs and the GRIMM were mounted on the east side of a small outbuilding.

**2.3 Data Analysis**
The measurements from the low-cost sensors and the research monitor (GRIMM) were converted to hourly average
concentrations and time-synchronized to MDT. Two EPA-flagged measurements corresponding to unexplainable high
hourly $PM_{10}$ concentrations (>800 µg/m³) from FEM-HW were removed. The low-cost sensors used in this study were
not supplemented with dryers, and therefore their performance is affected by high humidity conditions, which can
result in condensation and droplet formation (Samad et al., 2021). Consequently, the measurements corresponding to
relative humidity greater than 85% were excluded from the study (<2% of total measurements).

Using the HW and EQ meteorological measurements, we defined dust events as periods with $PM_{10}$ concentrations
exceeding 100 µg/m³ accompanied by winds exceeding 5 m/s at either site. These high winds were either observed at
the beginning or during dust events. Each dust event typically included a period of time when $PM_{10}$ concentrations
began increasing before reaching peak values. After wind speeds began to decrease, $PM_{10}$ concentration decreased
gradually. The dust events in this study included the entire time period when wind/$PM_{10}$ levels decreased until $PM_{10}$
concentrations reached background levels (<50 µg/m³). Table 2 (for HW) and Table 1S (for EQ) provide the





meteorological parameters (wind speed, wind direction, temperature, and RH), $PM_{2.5}$ and $PM_{10}$ concentrations, and
$PM_{2.5}/PM_{10}$ ratios for each event.

We performed a linear regression to relate the $PM_{10}$ concentration measurements of the low-cost sensors to reference
monitors at HW and EQ and a research monitor at the RS. Performance guidelines for low-cost $PM_{10}$ measurements
are not yet available. For discussion purposes, we use EPA guidelines for low-cost $PM_{2.5}$ sensors, which include
acceptable performance as a slope of $1 \pm 0.35$, intercept of $0 \pm 5$ µg/m$^3$, root mean square error (RMSE) $\leq 7$ µg/m$^3$,
normalized root mean square error (NRMSE) $\leq 30\%$, and $R^2 > 0.7$ (when compared with the reference monitor)
(Rachelle M. Duvall et al., 2021). RMSE and NRMSE were calculated using the following equations:
$$RMSE = \sqrt{\frac{1}{N}\sum_{t=1}^{N}(low\ cost\ sensor_t - Ref_t)^2}$$

$$NRMSE = \frac{RMSE}{\overline{Ref}} \times 100$$

where, $low\ cost\ sensor$ represents the low-cost sensor measurement, $\overline{Ref}$ represents the reference/regulatory
measurements, and $\overline{Ref}$ represents the average of the reference or regulatory monitor measurements.

We also explored a $PM_{2.5}/PM_{10}$ ratio-based calibration strategy for correcting PMS sensor readings. Based on the ratio
of FEM-HW $PM_{2.5}/PM_{10}$, we segregated the FEM-HW and PMS-HW $PM_{10}$ measurements into six bins: $PM_{2.5}/PM_{10}$:
<0.2, 0.2-0.3, 0.3-0.4, 0.4-0.5, 0.5-0.7, and >0.7. For each bin, the co-located PMS-HW $PM_{10}$ concentrations were
linearly regressed against the FEM-HW $PM_{10}$ concentrations to obtain correction factors (slope and intercept). These
correction factors were later used to correct the PMS $PM_{10}$ concentrations at the other two locations (RS and EQ). The
$PM_{2.5}/PM_{10}$ ratios from the GRIMM and OPC-RS at the RS were calculated for use in the in selecting the appropriate
PM-ratio-based correction factor and subsequent correction of the collocated PMS sensors. At the EQ site, the
$PM_{2.5}/PM_{10}$ ratio from the FEM-EQ was used to select the appropriate PM-ratio-based correction factor and
subsequent correction of the nearby PMS sensors.



**Table 2:** Meteorological and PM characteristics during the dust events at the HW monitoring site. The number in the parenthesis
represents the minimum and maximum of the parameter. Parameters for the EQ site can be found in Table S1 (supplementary
material).

| Start | Duration (hr) | Wind Speed (m/s) | Relative humidity % | Temperature (ºC) | PM$_{2.5}$/PM$_{10}$ | PM$_{10}$ (µg/m$^3$) |
|---|---|---|---|---|---|---|
| 4/9/22 5:00 AM | 7 | 3.13 [1.13, 4.16]* | 37.9 [28, 46] | 10.4 [8.3, 13.8] | 0.14 [0.10, 0.27] | 81.3 [36, 140] |
| 4/11/22 10:00 AM | 9 | 4.12 [2.11, 5.91] | 20.9 [12, 37] | 12.4 [7.2, 15.6] | 0.2 [0.13, 0.36] | 67.6 [44, 101] |
| 4/19/22 9:00 AM | 10 | 3.75 [1.64, 5.60] | 23.4 [17,32] | 16.7 [13.3, 18.3] | 0.24 [0.13, 0.36] | 96.5 [54, 161] |
| 4/21/22 11:00 AM | 23 | 3.54 [1.02, 6.73] | 37.6 [10, 79] | 15.6 [7.2,23.9] | 0.15 [0.08, 0.24] | 141 [51, 274] |
| 4/28/22 9:00 PM | 4 | 3.17 [1.54, 5.14] | 36.5 [28, 45] | 14.4 [11.1, 17.2] | 0.2 [0.10, 0.38] | 79.5 [26, 128] |

*a wind speed of 6.27 m/s was observed at the EQ site

## 3 Results and Discussion

Figure 2 shows the hourly average PM$_{10}$ concentration at the three different sites, with the dust events highlighted in
grey. The five dust events were observed at all three locations, and they occurred at approximately the same time.
Four of the dust events lasted less than 10 hours, and the event on 21 April 2022 lasted 23 hours. The PM$_{2.5}$/PM$_{10}$ ratio
(Table 1) remained less than 0.3 during all the events, indicating the predominant contribution of coarser particles to
PM$_{10}$. For each event, the PM$_{10}$ concentrations reached at least 100 µg/m$^3$. During the 21$^{st}$ April event, hourly average
PM$_{10}$ concentrations reached 275 µg/m$^3$ at HW, 311 µg/m$^3$ at EQ, and 173 µg/m$^3$ at the RS site (Table 1 and Table
1S). The lower PM$_{10}$ concentration at the RS may be due to its residential location, its higher altitude, and its greater
distance from dust sources. The OPC-HW and OPC-RS PM$_{10}$ concentration estimates followed the temporal pattern
of the reference/research monitors including during the dust events. Previous studies have observed similar response
for OPC-N3 and OPC-N2 (previous version of the OPC-N3) for dust events (Masic et al., 2020; Vogt et al., 2021).
Vogt et al. (2021) found that the OPC-N3 tracked PM$_{10}$ concentrations from a FIDAS (EN 16450 approved regulatory
instrument) for long-range transport dust events (PM$_{10}$ range 60 – 125 µg/m$^3$). The PMS sensors followed the temporal
pattern of the reference/research monitors except during the dust events when the PMS sensors substantially
underestimated PM$_{10}$ concentration (Fig. 2). Vogt et al. (2021) also found that the PMS5003 underestimated the PM$_{10}$
concentration during dust events. In addition, Masic et al. (2020) reported that during the Aralkum Desert dust event
(PM$_{10}$ reached 160 µg/m$^3$), the PM$_{10}$ reported by OPC-N2 agreed well with the GRIMM 11-D (research-grade optical
particle sizer), whereas the PMS5003 was not able to detect a large fraction of coarse particles correctly. Most of these



studies recorded one dust event during their sampling duration, whereas this study found that the OPC-N3 tracked
multiple dust events.

Figure 3 shows wind roses for April 2022 and each of the dust events. During the month of April, winds exceeding 5
m/s were observed at HW during 2.5% of the hours (1.81 % south predominant and 0.69% west predominant). For
dust events observed on 11[th] April and 21-22[nd] April, the high winds came from the south, whereas, for the rest of the
events, high winds predominantly came from the west. The different wind directions could be transporting dust from
different sources, such as the playas to the south and west of the Salt Lake Valley, the exposed playas of the Great
Salt Lake, or local sources, such as mine tailing, gravel operations, unpaved roads, and an open-pit copper mine
(Hahnenberger and Nicoll, 2012; Perry et al., 2019). All study monitoring sites are located west and southwest of the
Great Salt Lake (Perry et al., 2019). Identifying the sources of the wind-blown dust and the effects of these differences
on sensor performance would require a thorough analysis of the meteorology, the PM composition, and size
distribution during the study period.

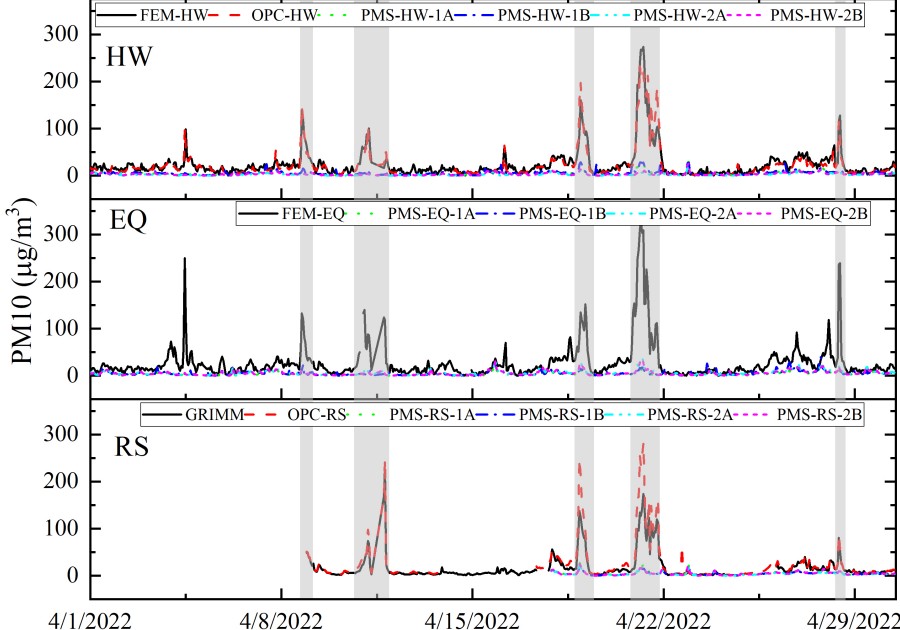


**Figure 2:** Hourly averaged $PM_{10}$ concentrations from the FEM, research monitors and low-cost sensors at the three different sites:
HW, EQ, and RS. Black solid lines represent reference/research monitors; red dash represents OPC-N3; green dot, blue dash-dot,
turquoise dash-dot-dot, and pink short-dash represent PMS sensors. The shaded peaks on 4/9/2022, 4/11/2022, 4/19/2022,
4/21/2022, and 4/28/2022 correspond to dust events. More details on these events can be found in Table 2.





## 3.1 OPC-N3 performance


Figure 4 illustrates the strong correlation between the OPC-N3 and the $PM_{10}$ concentration measured by the FEM at
the HW site and the GRIMM monitor at the RS where the coefficient of determination ranges from 0.865 and 0.937.
The intercept, slope, and $R^2$ were within the guidelines suggested by the EPA for low-cost $PM_{2.5}$ sensors, although the
RMSE and NRMSE (uncorrected measurements) exceeded the guidelines, 12.4 µg/m$^3$ and 53.5 %, respectively (Fig.
4). Vogt et al. (2021) also observed a similar slope (0.84-0.9 µg/m$^3$) and RMSE (12-13 µg/m$^3$) for OPC-N3 hourly
$PM_{10}$ compared to FIDAS, but with a lower correlation ($R^2$ 0.58-0.64) and for lower concentrations than this study.
Vogt et al. (2021) did not correct the $PM_{10}$ measurements for relative humidity, and approximately 20—30% of their
measurements corresponded to high humidity conditions (RH >85%), and the inclusion of elevated RH conditions
may have affected their correlations. The coefficient of determination in this study dropped to 0.81 after the inclusion
of measurements corresponding to RH above 85%, which corresponded to just 2% of the total measurements (Fig.
S1). Mukherjee et al. (2017) also reported correlations as high as 0.81 for OPC-N2 compared to BAM $PM_{10}$
measurements in the Cuyama Valley of California, with OPC-N2 reporting $PM_{10}$ concentrations of as high as 750
µg/m$^3$. Mukherjee et al. (2017) also did not correct the OPC data for relative humidity, which may have affected their
correlations. Our study as well as previous studies suggest that the OPC-N3/OPC-N2 tends to underestimate the $PM_{10}$
concentrations compared to the BAM (Mukherjee et al., 2017; Imami et al., 2022). The operating principle of the
BAM and OPC-N3 differ. The BAM $PM_{10}$ measurements are based on beta attenuation and do not require assumptions
about particle properties or particle size distribution. In contrast, OPCs rely on the measured particle size distribution
and assumed or measured particle properties (i.e., refractive index, shape, and density that can be size dependent) to
estimate mass concentration. In addition, particles < 0.3 µm in diameter do not scatter light sufficiently. Consequently,
some deviation from the mass measured by the FEM is expected. The assumptions about refractive index and shape
affect how particles are size classified, and in addition assumptions about density, affect estimates of mass
concentration.



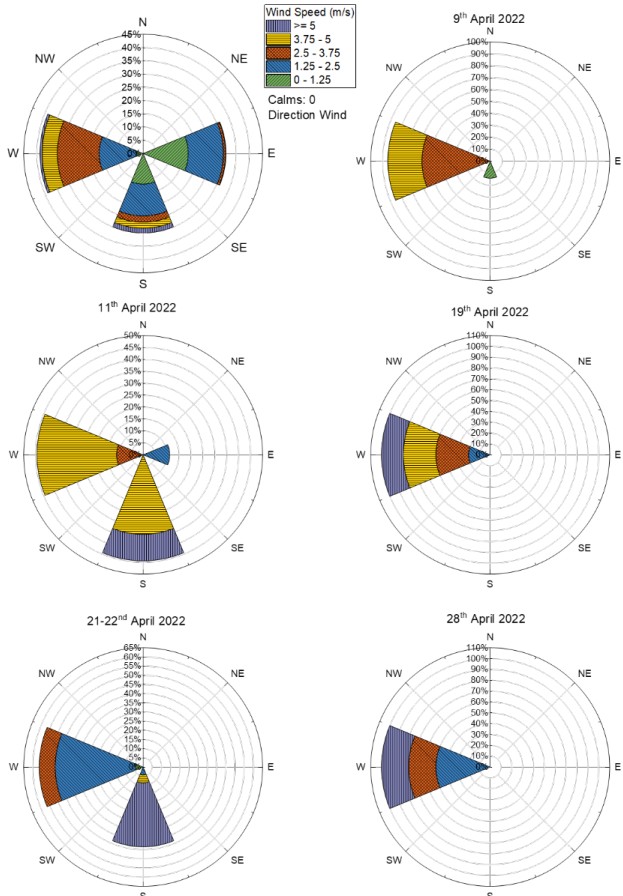

**Figure 3:** Wind roses for April 2022 and individual dust events, observed at HW. The wind roses for the EQ site can be found in the supplementary material (Fig. S13).

At the RS site, the OPC-RS showed a strong correlation with the GRIMM ($R^2$>0.9) and somewhat overestimated the $PM_{10}$ concentration (slope =1.45) compared to the GRIMM's default settings (Fig. 4). Such behavior from OPC-N3 and its predecessor model OPC-N2 has been observed previously. Crilley et al. (2018) also observed this same behavior for $PM_{10}$ for the OPC-N2 versus the GRIMM (1.108) and reported that the OPC-N2 estimated two to five times greater $PM_{10}$ mass than the GRIMM. Sousan et al. (2016) observed a slope of 1.6 for the Alphasense OPC-N2 compared to a GRIMM (1.108) for Arizona Road Dust. They attributed this behavior to the higher detection efficiency of OPC-N2 for particles > 0.8 μm compared to the GRIMM, and the effect of aerosol composition on OPC-N2 readings. Unlike Sousan et al. (2016), Bezantakos et al. (2018), using polystyrene spheres (size: 0.8, 1, 2.5, 5.1, 7.2, and 10.2 μm), reported that the OPC-N2 overestimated particle number concentrations, compared to GRIMM (1.109), for all sizes, not just size >1 μm.


Crilley et al.(2018) considered high relative humidity as a controlling factor behind the overestimation by the OPC-
N2. Badura et al. (2018) also reported a strong effect of relative humidity on the OPC-N2 measurements. We excluded
measurement corresponding to RH > 85% because we focus on dust events, and RH is low during these events. We
investigated the effect of RH (after excluding values > 85%) by performing a multilinear regression with the FEM-
HW as the dependent variable and the OPC-HW $PM_{10}$ concentration and RH as independent variables. Adding RH
did not significantly improve the correlation coefficient (not including RH: $R^2 = 0.865$, RMSE = 12 µg/m³; including
RH: $R^2 = 0.872$, RMSE = 11.7 µg/m³; Section S1, Supplementary material). Hygroscopic growth changes with PM
composition (Masic et al. 2020), and correcting measurements using a constant humidity coefficient can inject noise
into the results. In addition, the Salt Lake Valley is in an arid region, and 82% of PM measurements corresponded to
an RH of less than 60%. Consequently, the measurements were not corrected for the relative humidity for this study.

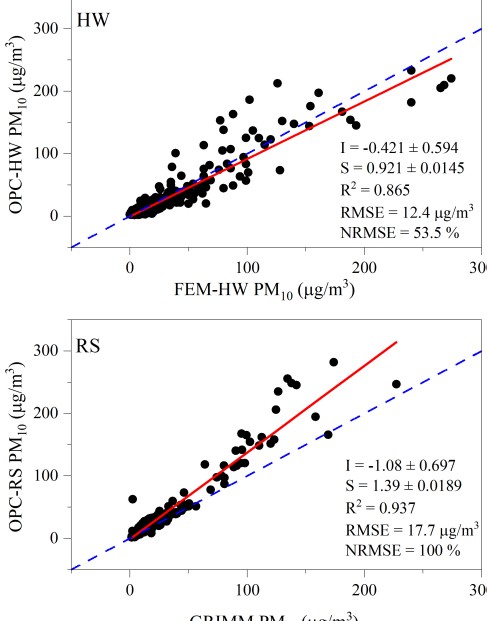


**Figure 4:** $PM_{10}$ concentration (top) OPC-HW vs. FEM-HW $PM_{10}$ concentration for the period between 04/1/2022-04/30/2022,
(bottom) OPC-RS vs. GRIMM $PM_{10}$ concentration at the RS for the sampling period 04/09/2022-04/30/2022. The red solid line
represents linear fit, and the blue dashed line represents the 1:1 line. I: intercept; S: slope.

**3.2 Performance of the PMS5003**
Figure 5, Figure 7 (top), and Figure 8 (top) illustrate the PMS sensors' poor-to-moderate correlations ($R^2$ between
0.128 and 0.482) with reference/research measurements of $PM_{10}$ concentration; these sensors underestimate the $PM_{10}$
concentration (slope < 0.09), particularly during dust events. These sensors also show high RMSEs (>30 µg/m³). Poor



performance of PMS sensors for PM$_{10}$ has been reported previously (Masic et al., 2020; Sayahi et al., 2019). Unlike
the OPC-N3, PMS sensors are nephelometers (Ouimette et al., 2022) and not optical particle counters, and their
response decreases with increasing size. Previous studies reported decreased response from PMS 5003 sensors for
particles larger than 0.5 μm (He et al., 2020; Kuula et al., 2020; Tryner et al., 2020). Kuula et al. (2020) and Tryner et
al. (2020) observed constant particle size distributions from the PMS 5003 regardless of actual particle size (exposed
monodisperse particles from polystyrene latex spheres, 0.1 – 2 μm, or generated with di-octyl sebacate 0.5– 10 μm).
The PMS sensors' inability to aspirate particles larger 2.5 μm is a significant cause of these sensors' inability to detect
coarse particles (aerodynamic size between 2.5 – 10 μm), such as those that dominate dust events (Ouimette et al.

362 2021).


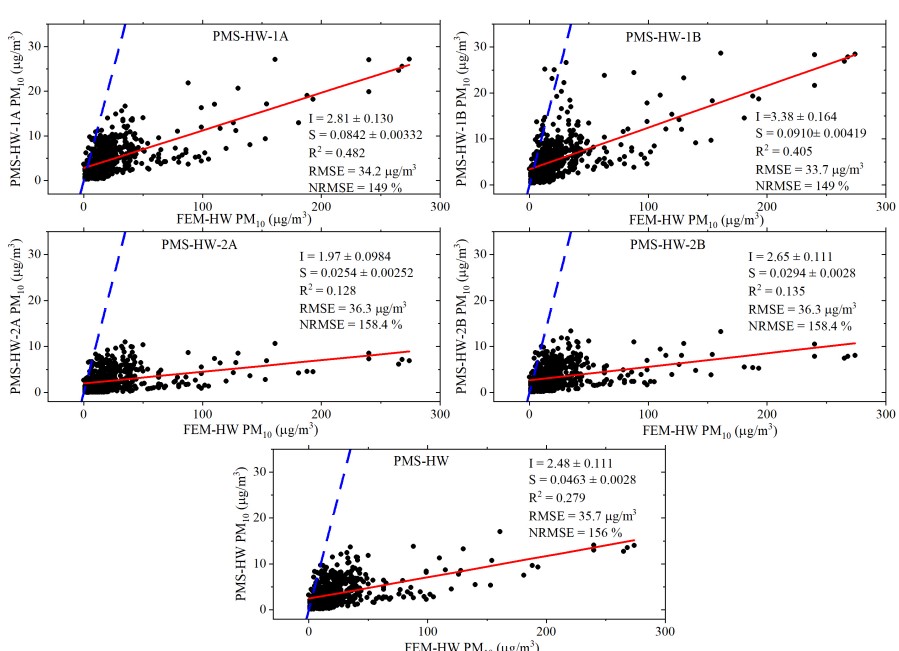


**Figure 5:** PMS PM$_{10}$ concentration vs. FEM-HW PM$_{10}$ concentration. PMS-HW represents the average of three PMS sensors
(PMS-HW-1A, PMS-HW-2A, and PMS-HW-2B). The red solid line represents linear fit, and the blue line represents the 1:1 line.
The plot includes measurements recorded between 04/1/2022 – 04/30/2022. I: intercept, and S: slope.


The PMS sensors also exhibited some inter-sensor variability during this study (Fig. S2). One sensor, PMS-HW-1B,
exhibited a fair correlation with the other three PMS sensors (R$^2$ = 0.53-0.55 with slopes differing by more than 50%).
The remaining three sensors (when compared to each other) had R$^2$ greater than 0.7, although their slopes differed by
40% (slope: PMS-HW-2A vs. PMS-HW-1A = 0.504; PMS-HW-2B vs PMS-HW-1A = 0.577). In terms of response

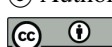



to $PM_{10}$ and correlation with the reference monitor, PMS-HW-1(A and B) performed somewhat better than PMS-HW-
2 (A and B) (RMSE < 35 μg/m³ and $R^2$ > 0.4, compared to RMSE < 36 and $R^2$ > 0.15).

Sensor-to-sensor variability has been reported in previous studies of PMS sensors, particularly for $PM_{2.5}$ concentration
(Sayahi et al., 2019; Tagle et al., 2020). The two PurpleAir II sensors (four PMS sensors) at the HW site were deployed
on different dates. PMS-HW-1 was deployed on 4/24/2020, whereas the PMS-HW-2 was deployed on 9/20/2019.
These sensors could be from different manufacturing batches, and they experienced different amounts of time in the
field. Sensor aging can cause differences in PMS sensor performance (Tryner et al., 2020). In addition, because the
PMS sensors are inefficient at measuring particles larger than $PM_{2.5}$ μm in diameter, as evidenced by the low slopes
in Figure 5, small differences (potentially due to sensor orientation and inherent differences in the sensors themselves)
can magnify sensor to sensor variability. Mukherjee et al. (2017) and Duvall et al. (2021) discuss the importance of
sampler positioning for $PM_{10}$ measurements.  For presentation purposes, we have excluded the PMS-HW-1B, which
exhibited poor correlation with the other PMS sensors (PMS-HW-1A, PMS-HW-2A, and PMS-HW-2B), and
averaged the remaining three PMS $PM_{10}$ concentrations at HW and compared the average of the three sensors to the
$PM_{10}$ concentrations measured by the FEM. Figure 5 shows the poor $R^2$ between the average of all PMS sensors and
FEM $PM_{10}$ (R2 = 0.279), and how the PMS-HW underestimates the $PM_{10}$ composition (slope of 0.0463).

**3.3 Using $PM_{2.5}$/$PM_{10}$ ratios to obtain size-segregated PMS correction factors**

The effect of correcting the PMS measurements with $PM_{2.5}$/$PM_{10}$ ratio-based factors on PMS performance was
explored as a strategy to obtain correction factors that could enable the PMS measurements to infer $PM_{10}$
concentrations. The $PM_{2.5}$/$PM_{10}$ ratio, calculated using the $PM_{2.5}$ and $PM_{10}$ concentrations reported by the FEM-HW,
was used to segregate the PMS-HW measurements into six bins: $PM_{2.5}$/$PM_{10}$: <0.2, 0.2-0.3, 0.3-0.4, 0.4-0.5, 0.5-0.7,
>0.7. For all the binned ratios (Figure 6), the PMS showed a consistent $R^2$ of greater than 0.6 (compared to $R^2$ values
of 0.128 – 0.482 prior to binning), but with very different slopes for the different $PM_{2.5}$/$PM_{10}$ bins. The slope varied
between 17 – 1.07, with the magnitude decreasing with the $PM_{2.5}$/$PM_{10}$ ratio. Note that Figures 4 and 5 show the FEM
on the x axes, whereas Figure 6 shows the regression equations used for correcting the PMS measurements (with FEM
on the y axes). During the dust events, the $PM_{2.5}$/$PM_{10}$ ratio was less than 0.3, supporting the large contribution from
dust and the corresponding large magnitude of $PM_{10}$ concentration. The $PM_{10}$ concentrations were lowest for the high
$PM_{2.5}$/$PM_{10}$ ratios (>0.7), and most $PM_{10}$ concentrations were below 5 μg/m³, which is close to the BAM's lower limit
of detection (Met One Technical Bulletin BAM-1020 Detection Limit, 2022) and likely contributes to the low
correlation observed for this ratio.

The slope and intercept for each bin were used as correction factors, called PM-ratio-based correction factors, to
correct the PMS $PM_{10}$ measurements at the other two locations, i.e., RS and EQ.

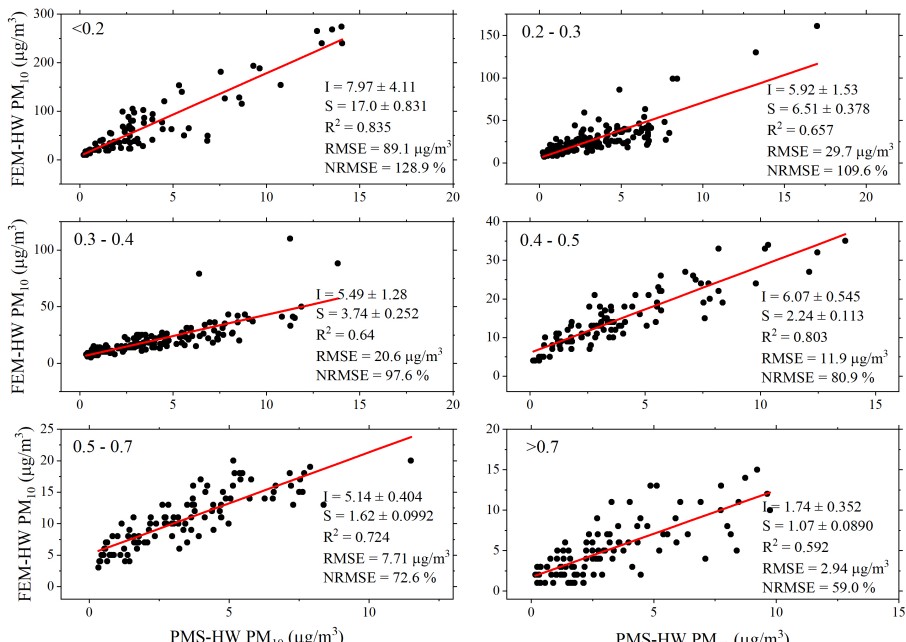

**Figure 6:** PMS-HW PM$_{10}$ concentration (average of three PMS sensors at HW) vs. FEM-HW PM$_{10}$ concentration for different PM$_{2.5}$/PM$_{10}$ bins. The RMSE and NRMSE has units µg/m$^3$ and %, respectively.

## 3.4 Correcting PMS data at RS and EQ sites

Similar to the HW site, the PMS PM$_{10}$ concentration measurements at the RS (Fig. 7, top) exhibited poor-to-moderate correlation (R$^2$ between 0.32-0.49, RMSE > 33 µg/m$^3$) compared to the research monitor and underestimated the PM$_{10}$ concentrations (slope <0.099). We corrected the raw PMS PM$_{10}$ concentration measurements using the PM-ratio-based correction factors obtained from the HW site and the PM$_{2.5}$/ PM$_{10}$ ratio from the GRIMM or the OPC to select a correction factor for each of the six PM$_{2.5}$/ PM$_{10}$ bins. Using the GRIMM provided ratios, Figure 7 (middle) shows that at the RS, after PM-ratio-based correction of the PM$_{10}$ measurements, the correlation for all the PMS sensors improved significantly (R$^2$ > 0.77) and the RMSEs decreased (< 18 µg/m$^3$). The R$^2$ varied between 0.773-0.810, and the slopes varied between 0.526-0.717. The intercept was a little higher (7-10 µg/m$^3$) than the EPA suggested guideline for low-cost PM$_{2.5}$ sensors. All the PMS sensors at RS were freshly deployed and were all mounted on the east side of a small building. These sensors exhibited good inter-sensor correlation (Fig. S4, R$^2$ > 0.97, slope > 0.77) and therefore exhibited very similar improvement all the sensors using the PM-ratio-based correction. The correlations between PMS PM$_{10}$ and GRIMM PM$_{10}$ concentrations were also good (R$^2$>0.7) when considering PM$_{10}$ < 50 µg/m$^3$



(Fig. S8 vs. Fig. S9), indicating that PM-ratio-based correction factors are applicable during more typical ambient
levels of $PM_{10}$ (without dust events).

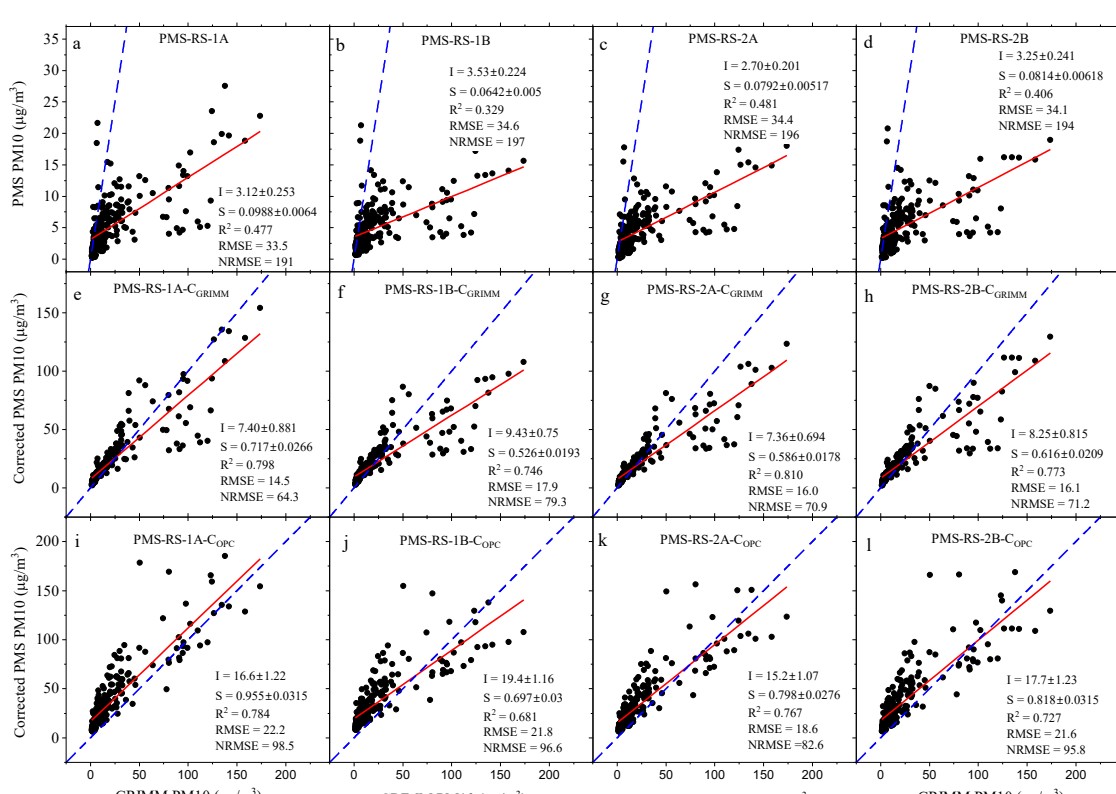



**Figure 7:** (Top: a, b, c, and d) Uncorrected PMS $PM_{10}$ concentration vs. GRIMM $PM_{10}$ concentration at RS the site. (Middle: e, f,
g, and h) Corrected $PM_{10}$ concentrations using the PM-ratio-based correction factors developed at HW and the $PM_{2.5}/PM_{10}$ ratios
provided by the GRIMM at the RS. (Bottom: i, j, k, and l) Corrected $PM_{10}$ concentrations using the PM-ratio-based correction
factors developed at HW and the $PM_{2.5}/PM_{10}$ ratios provided by the OPC-RS at the RS. The solid red line represents the linear fit
and the blue dash line represents the 1:1 line. The plots include measurements recorded between 04/18/2022 – 04/30/2022. I:
intercept; S: slope. The RMSE and NRMSE has units $\mu g/m^3$ and %, respectively.

Figure 7 (bottom) illustrates a similar strategy at the RS site but using the OPC-RS to provide the $PM_{2.5}/PM_{10}$ ratio. It
also shows that the correlation for PMS sensors improved after applying the PM-ratio-based correction using the OPC-
RS for the ratio ($R^2$ = 0.681 - 0.784). After correction, the slope also increased and varied between 0.589-0.813. The
corrected RMSE (18.6 – 22.2 $\mu g/m^3$) and intercept (15.2-19.4 $\mu g/m^3$) were somewhat higher than that observed when
using GRIMM-reported PM ratios (Fig. 7 (middle)). From Figure 7 (bottom), the PM-ratio-based corrected PMS $PM_{10}$
concentration for $PM_{10} < 50 \mu g/m^3$ was always above the 1:1 line, i.e., the PMS $PM_{10}$ concentration was overestimated.
The OPC-RS efficiency in counting particles smaller than 0.8 $\mu m$ is lower than the GRIMM (Bezantakos et al., 2018;
Sousan et al., 2016), and therefore underestimates $PM_{2.5}$ mass. Figure S5 also illustrates this overestimation in our





study, where for low $PM_{2.5}$ and $PM_{10}$ concentrations (90% of the measurements when $PM_{2.5}$ < 12 µg/m$^3$ and $PM_{10}$ <

40 µg/m$^3$) the OPC-RS underestimated the $PM_{2.5}$ mass compared to the GRIMM, although the OPC-RS $PM_{10}$

concentrations were similar to those of the GRIMM. The underestimated $PM_{2.5}$ measurements from the OPC affected

the $PM_{2.5}/PM_{10}$ ratios, which for the OPC-RS remained lower than those reported by the GRIMM (Fig. S6). The

magnitude of the PM-ratio-based correction factors (Fig. 6) was inversely related to the $PM_{2.5}/PM_{10}$ ratio. Since the

OPC-RS reported ratios were always low, the corrected $PM_{10}$ measurements below 50 µg/m$^3$ were overestimated (Fig

S10).

At the EQ site, we used the $PM_{2.5}/PM_{10}$ ratios from FEM measurements at the EQ site coupled with the PM-ratio-

based correction factors developed at the HW site to correct the PMS $PM_{10}$ concentrations from sensors located near

the EQ site. Correcting the PMS $PM_{10}$ concentrations using this approach did improve the correlation with FEM-EQ

(Fig. 8). Before the correction, all the PMS sensors has poor correlation with the FEM ($R^2$ < 0.342 and slope < 0.0737).

The $R^2$ improved to 0.617 - 0.797, and the slope increased to 0.602-1.38 after PM-ratio-based correction. The RMSE

decreased and ranged between 21.5 – 35.6 µg/m$^3$. The intercept increased and varied between 6.06-15.4. The sensors

at this site showed moderate inter-sensor correlation (Fig. S7), which was expected as these sensors were not

collocated. The different correlations with respect to FEM-EQ for the two PurpleAir II were also expected as these

sensors were not collocated with the FEM-EQ.

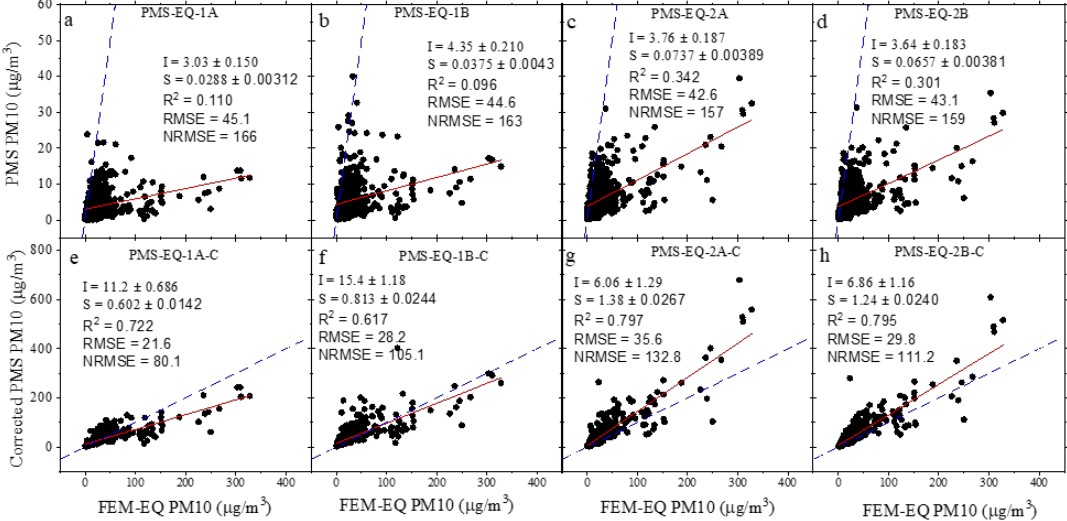

**Figure 8:** (top: a, b, c, and d) Uncorrected PMS $PM_{10}$ concentration vs. FEM-EQ $PM_{10}$ concentrations at the EQ site. (bottom: e, f, g, and h) Corrected $PM_{10}$ concentrations using the correction factors developed at HW and the $PM_{2.5}/PM_{10}$ ratios calculated using FEM-EQ $PM_{10}$ and $PM_{2.5}$ concentrations. The solid red line represents the linear fit and the blue dash line represents the 1:1 line. The plots include measurements recorded between 04/1/2022 – 04/30/2022. I: intercept; S: slope. The RMSE and NRMSE has units µg/m$^3$and %, respectively.


**4 Limitations**

This study has several limitations. The sensor's performance was evaluated for a month-long period in April 2022 and focused primarily on dust events, which commonly occur during this month. Understanding the OPC-N3 performance and whether using a $PM_{2.5}/PM_{10}$ ratio-based correction could improve correction factors for PMS sensors in other seasons and under different environmental conditions, like, wildfires, cold air pools, etc., would require a longer period of evaluation. This study used four PMS5003 sensors at the HW site and unlike the RS site, the sensors at HW were deployed at different times. These sensors showed moderate inter-sensor correlation, suggesting the need for further investigation of sensor age, sensor siting for $PM_{10}$ measurements, and potentially recalibration. This study occurred in an arid region, with RH generally less than 60%. This study did not find a significant improvement by adding RH to a calibration model between the OPC-N3 and the FEM. However, the applicability of this study's results to other, more humid, regions would need to be evaluated. The correction factors derived in this study used an average of three co-located PMS sensor measurements at a single site. In absence of detailed information about ambient particle properties, this study used default constant density for all the size-bins for OPC-N3. The Alphasense OPC-N3 allows the user to change the size-bin specific density for better estimates of $PM_{10}$, and if size-bin density and refractive index were available, the OPC measurements could potentially be improved. Our proposed PM-ratio-based calibration method relies on local measurements of the $PM_{2.5}/PM_{10}$ ratio. This requires FEM or other accurate measurements of $PM_{2.5}$ and $PM_{10}$ concentration, and the needed spatial distribution of these accurate $PM_{2.5}$ and $PM_{10}$ concentrations would need to be determined.

**5 Conclusions**

This study evaluated the performance of Alphasense OPC-N3 $PM_{10}$ measurements compared to FEM and GRIMM measurements during multiple dust events at two locations (HW and RS). The OPC-N3 tracked all the dust events at the two locations and exhibited a strong correlation with reference measurements ($R^2 = 0.865 - 0.937$), RMSE of 12.4-17.7 µg/m³, and NRMSE of 53.5 – 100 %. Uncorrected PMS5003 $PM_{10}$ measurements showed poor to moderate correlation ($R^2 < 0.49$) with the reference/research monitors at three locations (HW, RS, and EQ), with a RMSE of 33-45 µg/m³ and a NRMSE of 145-197 %. The PMS measurements severely underestimated the $PM_{10}$ concentrations (slope <0.099). We evaluated a PM-ratio-based correction method to improve estimates of $PM_{10}$ concentration from PMS sensors. After applying this method, PMS $PM_{10}$ concentrations correlated reasonably well with FEM measurements ($R^2 > 0.63$) and GRIMM measurements ($R^2 > 0.76$), the RMSE decreased to 15-25 µg/m³ and NRMSE decreased to 64 – 132 %. Our results suggest that it may be possible to leverage measurements from existing networks relying on low-cost $PM_{2.5}$ sensors to obtain better resolved spatial estimates of $PM_{10}$ concentration using a combination of PMS sensors and measurements of $PM_{2.5}$ and $PM_{10}$, such as those provided by FEMs, research-grade instrumentation, or the OPC-N3.


**Data Availability:**


The raw and processed data used in the manuscript can be found at: https://doi.org/10.7278/S50d-xbns-3ge3

**Authors Contribution:**


KEK and KK conceptualized the research, collected, and analysed the data. KK developed the original draft and KEK
reviewed the original draft. KK provided the supervision and acquired the funding.

**Competing interests:**


Dr. Kerry Kelly has a financial interest in the company Tellus Networked Solutions, LCC, which commercializes
solutions for environmental monitoring. Their technology was not used as part of this work.

**Acknowledgements:**


This material is based upon work supported by the National Science Foundation under Grant No. 2012091
Collaborative Research Network Cluster: Dust in the Critical Zone and under Grant No. 2228600  CIVIC-PG:
TRACK A: Community Resilience through Engaging, Actionable, Timely, High-Resolution Air Quality
Information (CREATE-AQI). Thanks to PurpleAir for donating two PAIIs to the project.

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

Karoline K. (Johnson) Barkjohn, Ian VonWal, Danny Greene, and Tim Dye: Performance Testing Protocols, Metrics,
and Target Values for Fine Particulate Matter Air Sensors: Use in Ambient, Outdoor, Fixed Site, Non-Regulatory
Supplemental and Informational Monitoring Applications, 2021.
Runström Eden, G., Tinnerberg, H., Rosell, L., Möller, R., Almstrand, A. C., and Bredberg, A.: Exploring Methods
for Surveillance of Occupational Exposure from Additive Manufacturing in Four Different Industrial Facilities, Ann
Work Expo Health, 66, 163–177, https://doi.org/10.1093/annweh/wxab070, 2022.



Samad, A., Mimiaga, F. E. M., Laquai, B., and Vogt, U.: Investigating a low-cost dryer designed for low-cost PM
sensors measuring ambient air quality, Sensors (Switzerland), 21, 1–18, https://doi.org/10.3390/s21030804, 2021.
Sayahi, T., Butterfield, A., and Kelly, K. E.: Long-term field evaluation of the Plantower PMS low-cost particulate
matter sensors, Environmental Pollution, 245, 932–940, https://doi.org/10.1016/j.envpol.2018.11.065, 2019.
Schweitzer, M. D., Calzadilla, A. S., Salamo, O., Sharifi, A., Kumar, N., Holt, G., Campos, M., and Mirsaeidi, M.:
Lung   health   in   era   of   climate   change   and   dust   storms,   Environ   Res,   163,   36–42,
https://doi.org/10.1016/j.envres.2018.02.001, 2018.
Sousan, S., Koehler, K., Hallett, L., and Peters, T. M.: Evaluation of the Alphasense optical particle counter (OPC-
N2) and the Grimm portable aerosol spectrometer (PAS-1.108), Aerosol Science and Technology, 50, 1352–1365,
https://doi.org/10.1080/02786826.2016.1232859, 2016.
Sousan, S., Regmi, S., and Park, Y. M.: Laboratory evaluation of low-cost optical particle counters for environmental
and occupational exposures, Sensors, 21, https://doi.org/10.3390/s21124146, 2021.
Soy, F. K.: The effects of dust storms on quality of life of allergic patients with or without asthma, The Turkish Journal
of Ear Nose and Throat, 26, 19–27, https://doi.org/10.5606/kbbihtisas.2016.56254, 2016.
Speranza, A., Caggiano, R., Margiotta, S., and Trippetta, S.: A novel approach to comparing simultaneous size-
segregated particulate matter (PM) concentration ratios by means of a dedicated triangular diagram using the Agri
Valley   PM   measurements   as   an   example,   Natural   Hazards   and   Earth   System   Sciences,   14,   2727–2733,
https://doi.org/10.5194/nhess-14-2727-2014, 2014.
Sugimoto, N., Shimizu, A., Matsui, I., and Nishikawa, M.: A method for estimating the fraction of mineral dust in
particulate   matter   using   PM2.5-to-PM10   ratios,   Particuology,   28,   114–120,
https://doi.org/10.1016/j.partic.2015.09.005, 2016.
Tagle, M., Rojas, F., Reyes, F., Vásquez, Y., Hallgren, F., Lindén, J., Kolev, D., Watne, Å. K., and Oyola, P.: Field
performance of a low-cost sensor in the monitoring of particulate matter in Santiago, Chile, Environ Monit Assess,
192, 171, https://doi.org/10.1007/s10661-020-8118-4, 2020.
Tam, W. W. S., Wong, T. W., Wong, A. H. S., and Hui, D. S. C.: Effect of dust storm events on daily emergency
admissions for respiratory diseases, Respirology, 17, 143–148, https://doi.org/10.1111/j.1440-1843.2011.02056.x,
705   2012.

Alphasense Ltd: https://www.alphasense.com/wp-content/uploads/2022/09/Alphasense_OPC-N3_datasheet.pdf, last
access: 12 October 2022.
Tong, D. Q., Wang, J. X. L., Gill, T. E., Lei, H., and Wang, B.: Intensified dust storm activity and Valley fever
infection   in   the   southwestern   United   States,   Geophys   Res   Lett,   44,   4304–4312,
https://doi.org/10.1002/2017GL073524, 2017.
Trianti, S.-M., Samoli, E., Rodopoulou, S., Katsouyanni, K., Papiris, S. A., and Karakatsani, A.: Desert dust outbreaks
and respiratory morbidity in Athens, Greece, Environmental Health, 16, 72, https://doi.org/10.1186/s12940-017-0281-
x, 2017.



Tryner, J., Mehaffy, J., Miller-Lionberg, D., and Volckens, J.: Effects of aerosol type and simulated aging on
performance of low-cost PM sensors, J Aerosol Sci, 150, 105654, https://doi.org/10.1016/j.jaerosci.2020.105654,
716    2020.
Vogt, M., Schneider, P., Castell, N., and Hamer, P.: Assessment of low-cost particulate matter sensor systems against
optical and gravimetric methods in a field co-location in norway, Atmosphere (Basel), 12,
https://doi.org/10.3390/atmos12080961, 2021.
Williams, A. P., Cook, B. I., and Smerdon, J. E.: Rapid intensification of the emerging southwestern North American
megadrought in 2020–2021, Nat Clim Chang, 12, 232–234, https://doi.org/10.1038/s41558-022-01290-z, 2022.
Xu, G., Jiao, L., Zhang, B., Zhao, S., Yuan, M., Gu, Y., Liu, J., and Tang, X.: Spatial and temporal variability of the
PM2.5/PM10    ratio    in    Wuhan,    Central    China,    Aerosol    Air    Qual    Res,    17,    741–751,
https://doi.org/10.4209/aaqr.2016.09.0406, 2017.