# Peer review of "Performance evaluation of the Alphasense OPC-N3 and Plantower PMS5003 sensor in measuring dust events in the Salt Lake Valley, Utah"

_Atmospheric Measurement Techniques, 2022_

## Author Response (AR2)

**Community**

A minor correction to your preprint - - on lines 61-63 it states that some low cost sensors are ineffective at measuring PM10 and dust, primarily due to the sensor's inability to aspirate these larger particles into the device, and cites our paper by Ouimette (2022). In our paper we did not state that poor aspiration efficiency was the primary reason for the coarse particle inefficiency. Based on the Plantower PMS5003 particular geometry our physical-optical model predicted an 80-90% reduction in the 2-um particle scattering coefficient compared to a perfect nephelometer, due to truncation of the forward scattering signal. However, the lab data showed more like a 95% reduction in 2-um light scattering. Based on Willeke and others' work we hypothesized that poor aspiration efficiency could be a reason for this additional 5-15% loss. But we neither modeled nor measured it for that paper. It's extremely difficult to measure aspiration losses for the PMS5003 because any measurement slows or stops its flow rate due to Δp across the little fan. I hope this helps. Best wishes on your paper review.

Response: We thank Dr. Ouimette for pointing this out. We have corrected the statement in the text to "……primarily due to truncation of the forward scattering coefficient for larger particles and in potentially due to the sensors' inability to aspirate the larger particles into the device"

**Refree 1**

Thank you for your answer to the comment by the public. I have read through the entire manuscript and it has been well-revised.

Response: We thank the referee for reviewing the manuscript.

**Refree 2**

Overall, this is a well written paper that explores the performance of 2 common air sensors for measuring PM10. This topic is of great interest to many air sensor users as many inexperienced users are using these devices without knowledge of their limitations for PM10. I have a number of minor comments and suggestions.

1. Multilinear regression with RH: Is this the correct form of equation to look at the influence of RH? It seems like past work with the Alphasense OPC has found nonlinear influence of RH (e.g., https://amt.copernicus.org/articles/11/709/2018/). How did you decide to use 85% as your cut-off?

Response: We thank the reviewer for pointing this out. We did follow the AMT article suggested by the reviewer for deciding the relative humidity cut point and the multi-linear regression. According to Crilley et al. (https://amt.copernicus.org/articles/11/709/2018/), the non-linear behavior (the exponential behavior) of the OPC-N2 with relative humidity was observed for relative humidity greater than 85%. They indicated that for relative humidity less than 85%, a linear response could be observed between then OPC response and relative humidity. This linear behavior for relative humidities below 85% was also evident from the figures in Crilley et al. article. Therefore we performed a multi-linear regression (excluding measurements with relative humidity > 85%) with FEM-HW PM10 as the dependent variable and OPC-HW (OPC at the Hawthorne monitoring station) and relative humidity (RH) as independent variables.

2. Limitations Line 480: I think it would be important to mention that the lack of significant improvement of adding RH was after excluding the high RH data.

We thank the reviewer for pointing this out. We have corrected the statement to include the relative humidity cut point as a limitation. Corrected statement: "However, this study excluded measurements with a RH > 85% (<2% of total measurements), a range where previous studies have identified a significant effect of RH (Crilley et al., 2018), and the applicability of this study's results to other, more humid, regions would need to be evaluated."

3. Lines 369-372: Did you consider if any of your PMS5003s were the "new" version of the PMS5003 and if that lead to any of the differences in agreement https://community.purpleair.com/t/new-version-of-plantower-pms5003/288/25, <a href="https://cfpub.epa.gov/si/si\_public\_record\_report.cfm?dirEntryId=355518&Lab=CEMM&simples">https://cfpub.epa.gov/si/si\_public\_record\_report.cfm?dirEntryId=355518&Lab=CEMM&simples</a> earch=0&showcriteria=2&sortby=pubDate&searchall=barkjohn&timstype=&datebeginpublished presented=02/03/2021

We thank the reviewer for pointing this out. We were aware of this issue, and we have confirmed with PurpleAir that all the PMS sensors used in this study were old versions.

 Did you consider whether the particle size bins provided by the PMS5003 provided any better estimate of PM10? (e.g., https://www.mdpi.com/1424-8220/22/13/4741, https://www.mdpi.com/1424-8220/22/7/2755, https://amt.copernicus.org/preprints/amt-2022-265/)

We did not attempt to calculate the PM mass from the numbers as we didn't know the density of the particles. Also, from our previous work and also reported by other researchers, the Plantower sensor does not respond to particles bigger than  $2.5 \mu m$ .

5. Table 2: It might be helpful to add a row for the non-dust impacted times for easy comparison. Same suggestion for Table S1.

We have added the non-dust times in both tables.

6. Consider adding averaging interval to all figure labels in text and SI (Assuming they are all 1-hr?)

We have added the statement describing the averaging interval, i.e., 1 hour, in all of our figures including supplementary material.

7. Fig S1: Coloring points with >85% RH would be helpful

We have marked the high humidity point as red in Fig. S1.

8. Introduction: Recommend adding the US EPA standard for PM 10

We have added the statement about the EPA standard for PM10 in the introduction. The statement added "The US EPA's national ambient air quality standard for  $PM_{10}$  concentration and are 150 and 50 µg/m3 for the 24-hour and annual average, respectively"

9. Typo in line 46

We have resolved the typo.